# PREPROCESSING ENHANCED IMAGE COMPRESSION FOR MACHINE VISION

## ABSTRACT

Recently, more and more images are compressed and sent to the back-end devices for machine analysis tasks (*e.g.,* object detection) instead of being purely watched by humans. However, most traditional or learned image codecs are designed to minimize the distortion of the human visual system without considering the increased demand from machine vision systems. In this work, we propose a preprocessing enhanced image compression method for machine vision tasks to address this challenge. Instead of relying on the learned image codecs for end-to-end optimization, our framework is built upon the traditional non-differential codecs, which means it is standard compatible and can be easily deployed in practical applications. Specifically, we propose a neural preprocessing module before the encoder to maintain the useful semantic information for the downstream tasks and suppress the irrelevant information for bitrate saving. Furthermore, our neural preprocessing module is quantization adaptive and can be used in different compression ratios. More importantly, to jointly optimize the preprocessing module with the downstream machine vision tasks, we introduce the proxy network for the traditional non-differential codecs in the back-propagation stage. We provide extensive experiments by evaluating our compression method for several representative downstream tasks with different backbone networks. Experimental results show our method achieves a better trade-off between the coding bitrate and the performance of the downstream machine vision tasks by saving about 20% bitrate.

## 1 INTRODUCTION

Due to the successful applications of deep neural networks, the machine vision tasks such as detection and classification have made a lot of progress in recent years (Tian et al., 2019; Ren et al., 2015; Lin et al., 2017; Redmon & Farhadi, 2018; Bochkovskiy et al., 2020; He et al., 2016; Zhang et al., 2020; Xie et al., 2017). Therefore, more and more images are captured by the front-end devices (*e.g.,* cameras) and sent to the back-end (*e.g.,* cloud servers) for machine analysis. According to the report from CISCO (2020), the percentage of the connections from this machine-to-machine scenario will be up to 50% in the future. Therefore, how to reduce the transmission bitrate while maintaining performance for the downstream machine vision tasks has become a challenge for the image compression field.

Unfortunately, although several traditional image compression standards, such as JPEG (Wallace, 1992) and BPG (Bellard, 2014), have been proposed in the past decades, they are designed to minimize the compression distortion for the human visual system (*e.g.*, PSNR) instead of the machine vision tasks (see Fig. 1(a)). More importantly, most compression standards are non-differential, which cannot be jointly optimized with the neural network based machine analysis methods. Therefore, the existing compression-then-analysis pipeline with the traditional codecs may not be optimal when we mainly focus on the performance of the downstream machine analysis. Recently, learned image compression methods (Ballé et al., 2017; 2018; Minnen et al., 2018) start to gain a lot of attention. Several approaches (Torfason et al., 2018; Akbari et al., 2019; Hu et al., 2020) also try to jointly optimize the learned compression methods with the downstream analysis tasks. However, the computational complexity for the learned image compression is usually high, and the standardization is not finalized; therefore, the massive deployment of learned compression approaches is

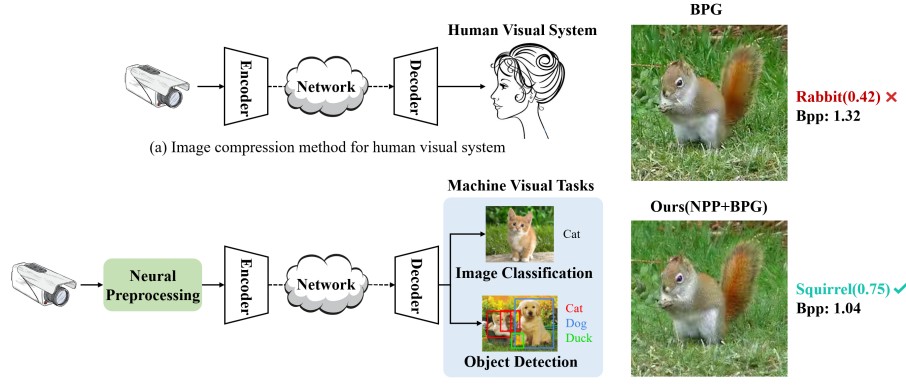

Figure 1: (a) Image compression method for human visual system. (b) Our proposed preprocessing enhanced image compression for machine vision tasks. (c) Image classification results for the image from the BPG codec and ours(NPP+BPG).

unlikely to happen soon, which means these approaches (Torfason et al., 2018; Akbari et al., 2019; Hu et al., 2020) may not be feasible in practical applications.

To address these challenges, we propose a preprocessing enhanced image compression framework for machine vision as shown in Fig. 1(b). Our framework builds upon the traditional standard compatible image codecs and can be easily applied to the practical compression-then-analysis systems. Specifically, we propose a neural preprocessing (NPP) module before the traditional codec and the input image will be filtered before encoding. After that, the decoded image is used for the downstream vision tasks, like detection or classification. To enable the end-to-end optimization, we further introduce the proxy network for the traditional non-differential image codecs (*e.g.,* BPG) in the training stage, where the gradients of the proxy network are propagated to the neural preprocessing module. Therefore, the proposed preprocessing module will be optimized to maintain the meaningful semantic information and reduce the irrelevant information for machine vision tasks, which leads to a better trade-off between the coding bitrate and machine analysis performances (see Fig. 1(c)). Furthermore, the proposed neural preprocessing module is quantization adaptive and can be integrated into traditional codecs with different compression ratios. To demonstrate the superiority of our preprocessing enhanced image compression method, we perform extensive experiments on several representative machine vision tasks (*e.g.,* object detection and image classification) with different downstream backbone networks. Experiments show that compared with the existing traditional codecs like BPG (Bellard, 2014), the proposed approach can save about 20% bitrate for the downstream tasks with the same accuracy.

The main contributions of our work are summarised as follows,

- Building upon the traditional codec, we propose a neural preprocessing module to generate the filtered images, which can be effectively compressed by the traditional codecs with high machine perception performance.

- To enable an end-to-end optimization for a better trade-off between coding bitrate and machine perception performance, we introduce the learned proxy network to approximate the traditional codecs for the back-propagation in the training stage.

- Experimental results show our approach is general and the optimized NPP model for one specific scenario can be used for other codecs, downstream backbones, or even other tasks.

## 2 RELATED WORKS

### 2.1 IMAGE COMPRESSION

Many traditional image compression algorithms (Wallace, 1992; Skodras et al., 2001; Bellard, 2014) have been proposed in the past decades. These methods are based on hand-craft techniques (*e.g.,* Discrete Cosine Transform) to reduce spatial redundancy. Recently, the learned image compression

methods (Ballé et al., 2017; Johnston et al., 2018; Toderici et al., 2017; Ballé et al., 2018; Minnen et al., 2018; Zhu et al., 2021; Cheng et al., 2020; Chen et al., 2021; Xie et al., 2021) have become popular. Latest works (Cheng et al., 2020; Chen et al., 2021; Xie et al., 2021; Zhu et al., 2021; He et al., 2021; 2022) also propose to use more powerful transform networks, such as residual blocks (Cheng et al., 2020), nonlocal layers (Chen et al., 2021), invertible layers (Xie et al., 2021) and transformer (Zhu et al., 2021). Although these approaches have achieved better compression performance, they are computationally expensive.

## 2.2 IMAGE COMPRESSION FOR MACHINE VISION

Most existing image compression methods (Wallace, 1992; Bellard, 2014; Ballé et al., 2017; 2018) aim to reduce reconstruction distortion in terms of the human visual system and are optimized based on pixel field metrics such as PSNR. With the development of deep learning, some studies began to explore connections between compression and downstream tasks (Gueguen et al., 2018; Ulicny & Dahyot, 2017; Ehrlich & Davis, 2019; Wang et al., 2022), some studies (Torfason et al., 2018; Akbari et al., 2019; Hu et al., 2020; Li et al., 2021; Wang et al., 2021; Yan et al., 2021; Fischer et al., 2021; Choi & Bajić, 2022; Özyılkan et al., 2023) also focus on the joint optimization of the image compression and the downstream machine vision tasks. For example, Torfason et al. (2018) proposed to directly perform image understanding tasks, such as classification and segmentation, on the compressed representations produced by the learning-based image compression methods.

Recently, a self-supervised learning scheme (Feng et al., 2022) is proposed to constrain the intermediate-layer features to be semantics-complete and achieved high performances in different downstream vision tasks. However, most existing works have to rely on the learning based codecs to enable the end-to-end optimization, which may not be feasible in the practical application considering the mainstream codecs are traditional ones. In contrast, our framework is built upon the traditional codecs and can also be end-to-end optimized through the proxy network.

## 2.3 PREPROCESSING

In the past decades, several methods (Xiang et al., 2016; Vidal et al., 2017; Doutre & Nasiopoulos, 2009) have been proposed to use the preprocessing technique to improve the performance of the image and video compression algorithms. Most of these methods are based on the Just Noticeable Distortion (JND) technique (Yang et al., 2005) and try to improve the perceptual quality of reconstructed frames. For example, Xiang et al. (2016) proposed adaptive perceptual preprocessing by removing information that is not perceptible to the human visual system. Vidal et al. (2017) combined several adaptive filters to denoise the image for bitrate saving.

In recent years, several learning-based preprocessing methods have been proposed (Chadha & Andreopoulos, 2021; Guleryuz et al., 2021; Talebi et al., 2021; Son et al., 2021). Chadha & Andreopoulos (2021) proposed a rate-aware perceptual preprocessing module for video coding. Guleryuz et al. (2021) proposed neural network based preprocessing and postprocessing modules to improve the compression performance of the traditional codecs. Talebi et al. (2021) designed a pre-editing neural network on the JPEG method to improve the visual quality of reconstructed images. Klopp et al. (2021) proposed to borrow the rate estimator of learning-based codec to boost the perceptual quality of tradictional codecs such as JPEG. We propose using the neural network based preprocessing method to improve the compression performance in machine vision instead of the human visual system.

# 3 PROPOSED METHOD

## 3.1 OVERVIEW

The overall architecture of our preprocessing enhanced image compression framework for machine vision is shown in Fig. 2. The whole system aims to achieve a better trade-off between coding bitrate and the performance of the machine analysis task. Specifically, we first feed the input image $X$ to the neural preprocessing module (NPP) for non-linear transform and generate the filtered image $\bar{X}$, which is expected to maintain the critical semantic information. Then, $\bar{X}$ is encoded and

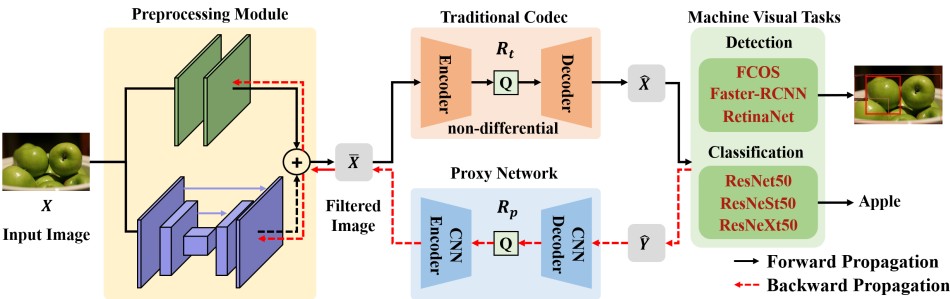

Figure 2: Overview of our preprocessing enhanced image compression for machine vision.

reconstructed by a traditional codec, like BPG (Bellard, 2014). Finally, the decoded $\hat{X}$ is input to machine analysis networks, such as FCOS (Tian et al., 2019).

Since the traditional codecs may not be differential, the proposed preprocessing module cannot enjoy the benefits of the joint end-to-end optimization with the downstream machine analysis tasks. To solve this problem, we additionally introduce a learned image compression network as the proxy network for the traditional codec in the training stage and the gradients of the proxy network are propagated to the preprocessing module (see Section 3.3 for more details). Here, we use BPG (Bellard, 2014) as the traditional codec in our implementation.

Then the framework is optimized by using the following loss function,

$$\mathcal{L} = R_t + \lambda \mathcal{D}_m + \beta D_{pre} \tag{1}$$

where $\mathcal{D}_m$ and $R_t$ represent the loss of the downstream machine vision task based on reconstructed image $\hat{X}$ and coding bitrate from the traditional codec, respectively. $\lambda$ is a hyper-parameter used to control the trade-off. In addition, to stabilize the training process, we also consider the distortion between the input image $X$ and the enhanced image $\bar{X}$, which is denoted as $D_{pre}$. $\beta$ is the constant weight parameter.

### 3.2 NEURAL PREPROCESSING NETWORK

As shown in Fig. 3, we provide the network architecture of our neural preprocessing module. Specifically, the original image $X$ is input into two parallel branches, where the first branch uses $1 \times 1$ convolutional layers to learn non-linear pixel-level transforms, and the second branch uses a U-Net (Ronneberger et al., 2015) style network to extract the semantic information. The outputs of two branches are added together as the final filtered image $\bar{X}$, which preserves the useful texture and semantical information through both shallow and deep transforms.

Furthermore, considering the traditional codecs have different compression ratios (*i.e.,* quantization parameter), the neural preprocessing module is required to generate the optimal filtered image $\bar{X}$ for each compression ratio. Here, we propose a quantization adaptation layer for the neural preprocessing module, which leads to an adaptive preprocessing based on the quantization parameters in the codec. As shown in Fig. 3, we integrate the quantization adaptive layer into the NPP module and scale the intermediate features for adaptive filtering. Specifically, based on the given quantization parameter (QP) in the traditional codec, we use a 2-layer MLP network to generate the scale vector $s$ and the output feature $f'$ is the channel-wise multiplication product between input feature $f$ and the generated scale vector $s$, *i.e.,* $f' = f \odot s$. Based on this strategy, the intermediate features in the preprocessing module will be modulated based on the quantization parameter; therefore, our module will generate the optimal filtered image $\bar{X}$ for the given QP in the BPG codec and achieve a better rate-accuracy trade-off.

Here we give an example in Fig. 4 to show the effectiveness of our preprocessing module. Fig. 4(a) and (b) represent the original image and output from the NPP module, respectively. Moreover, the corresponding compressed file sizes using the BPG (Bellard, 2014) ($QP = 37$) codec are 63.7kb and 47.0kb. At the same time, Fig. 4(c) shows that the information discarded by the preprocessing module is mainly distributed in the background region. In contrast, based on the GradCAM method (Selvaraju et al., 2017), the classification network (He et al., 2016) focuses on the foreground *Dingo* in the image as shown in Fig. 4(d,e). Simultaneously, (d) and (e) reveal that the classification model

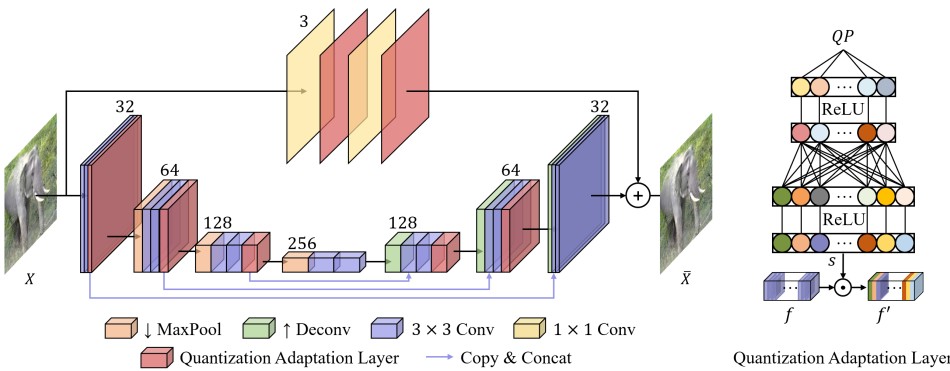

Figure 3: The implementation of our neural preprocessing module. The numbers represent the numbers of output channel for different operations.

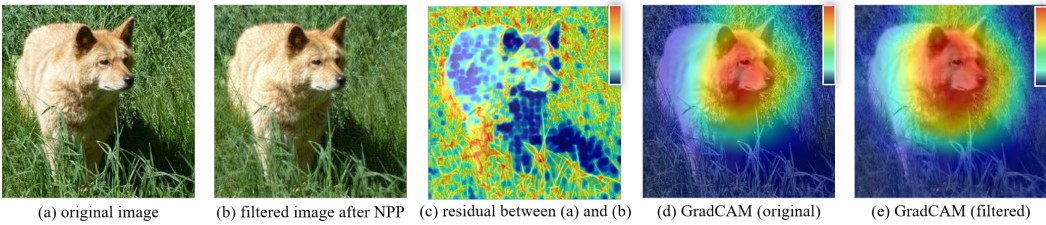

(a) original image    (b) filtered image after NPP    (c) residual between (a) and (b)    (d) GradCAM (original)    (e) GradCAM (filtered)

Figure 4: Visualization results of the neural preprocessing module. (a) is the original image. (b) is the filtered image after NPP. The color in (c), (d) and (e) represents the values of residual and localized class-discriminative regions form GradCAM(ResNet) (Selvaraju et al., 2017), where larger values are denoted by red color.

focuses more extensively on the expected region of the Dingo in the preprocessed image, potentially leading to a more accurate classification outcome. We present more illustrations in the Appendix. These results prove that the preprocessing module can preserve more critical semantic information for the downstream analysis tasks and reduce the irrelevant information for bitrate saving.

## 3.3 PROXY NETWORK

In our framework, to enable an end-to-end optimization for the whole system, a learned image compression network is introduced as the proxy network to replace the traditional codec during the backward propagation stage. Here, we use Minnen's approach (Minnen et al., 2018) as our proxy network.

To make sure that the proxy network can well approximate the traditional codec, the reconstruction quality of the BPG and Minnen's approach should be similar. The learned image compression approach (Minnen et al., 2018) is optimized based on Rate-Distortion (R-D) loss $R + \lambda_p D$ and the quality of the reconstructed image depends on the hyper-parameter $\lambda_p$. We start by selecting a pretrained image compression model, which, after being optimized with the R-D distortion loss and a suitable $\lambda_p$ parameter, has a performance comparable to the BPG. To make it even more aligned with BPG, we then fine-tune the proxy network using the following approach,

$$\mathcal{L}_p = R_p + \lambda_p D_p = R_p + \lambda_p d(\hat{X}, \hat{Y}) \tag{2}$$

where $d(\hat{X}, \hat{Y})$ denotes the distortion between the reconstructed image $\hat{X}$ from the BPG and the reconstructed image $\hat{Y}$ from the proxy network (see Fig. 2). $R_p$ represents the corresponding bitrate from the proxy network. After that, we obtain an optimized proxy codec to mimic the BPG codec.

We provide more implementation details of the end-to-end training procedure in Algorithm 1. In forward propagation, we can get the processed image $\bar{X}$ based on the input image $X$, where $\theta_{pre}$ represents the parameters of preprocessing module. Then the processed image is compressed by BPG codec and BPG will calculate the bitrate $R_t$ and produce the reconstructed image $\hat{X}$. At the

---

**Algorithm 1** Training Procedure

---

**Require:** training dataset $D = \{(X_k)\}_{k=1}^m$, trade-off parameter $\lambda, \beta = 0.5$, learning rate $\eta$

1: **for** all $(X) \in D$ **do**
2:      $\bar{X} \leftarrow Preprocessor(X|\theta_{pre})$     ▷ $\theta_{pre}$ represents the parameters of preprocessing module
3:      $\hat{X}, R_t \leftarrow BPG(\bar{X})$
4:      $\hat{Y}, R_p \leftarrow ProxyNetwork(\bar{X})$
5:      $\hat{Y}.data \leftarrow \hat{X}.data$
6:      $R_p.data \leftarrow R_t.data$
7:      $\hat{O} \leftarrow detector(\hat{Y})$
8:      $L \leftarrow R_p + \lambda \cdot loss(\hat{O}) + \beta \cdot mse(X, \bar{X})$                   ▷ calculate loss
9:      $g_{task} \leftarrow \dfrac{\partial L}{\partial \hat{O}} \cdot \dfrac{\partial \hat{O}}{\partial \hat{Y}}$               ▷ calculate gradients of $detector$
10:     $g_p \leftarrow g_{task} \cdot \dfrac{\partial \hat{Y}}{\partial \bar{X}} + \dfrac{\partial L}{\partial R_p} \cdot \dfrac{\partial R_p}{\partial \bar{X}}$       ▷ calculate gradients of $ProxyNetwork$
11:     $g_{pre} \leftarrow g_p \cdot \dfrac{\partial \bar{X}}{\partial X} + \dfrac{\partial \beta \cdot mse(X, \bar{X})}{\partial \bar{X}} \cdot \dfrac{\partial \bar{X}}{\partial X}$     ▷ calculate gradients of $Preprocessor$
12:     $\theta_{pre} \leftarrow \theta_{pre} - \eta \cdot g_{pre}$                ▷ optimize parameters of $Preprocessor$
13: **end for**

---

same time, we also generate the corresponding reconstructed image $\hat{Y}$ and bitrate $R_p$ based on the proxy network. Here, the values of $\hat{Y}$ and $R_p$ will be reassigned to $\hat{X}$ and $R_t$ from the BPG (Bellard, 2014) codec as shown in Line 5 and 6 in Algorithm 1. Then the reassigned $\hat{Y}$ is input to the analysis models (*e.g.*, an object detection module) and used to calculate the machine perception loss $\mathcal{D}_m$. After that, we can calculate the loss function as shown in Line 8. Based on this operation, we can use the bitrate and reconstructed image from the BPG in forward propagation and calculate the value of loss function while using the gradients of the proxy network in backward propagation. Finally, we perform the backward propagation, and optimize the weights of the neural preprocessing module, as shown in Line 9-12 in Algorithm 1. In the backward propagation, the gradients of the machine vision task model, proxy network, and preprocessing module, denoted as $g_{task}$, $g_p$, and $g_{pre}$, will be calculated sequentially. And the weight of preprocessing module will be optimized while the weights of other modules are fixed.

## 4 EXPERIMENTS

### 4.1 EXPERIMENTAL SETUP

**Datasets, Backbone Models and Evaluation Metrics.** For the object detection task, we train our framework on the COCO 2017 training set (Lin et al., 2014). The mean average precision (mAP) results are reported by evaluating the proposed framework on the COCO 2017 validation set which contains 5k images. In our experiments, three object detection baselines, FCOS (Tian et al., 2019), Faster-RCNN (Ren et al., 2015) and RetinaNet (Lin et al., 2017) are used for evaluation.

For the image classification task, we use the ImageNet dataset (Deng et al., 2009). The Top-1 accuracy is reported in our experiments. To demonstrate the effectiveness of our approach, we use ResNet (He et al., 2016), ResNeSt (Zhang et al., 2020) and ResNeXt (Xie et al., 2017) for the performance evaluation.

To showcase the inter-task generalization ability of our proposed NPP module, we also evaluate our approach for the semantic segmentation task and pose estimation tasks and the corresponding backbone networks are DeepLabv3 (Chen et al., 2017) and Deeppose (Toshev & Szegedy, 2014).

We further evaluate the compression performance in terms of the human visual system by using the perceptual metrics LPIPS (Zhang et al., 2018) on the Kodak dataset (Toderici et al., 2017). We use bits-per-pixel(bpp) in all experiments to measure the coding cost during the compression procedure.

**Implementation Details.** Our whole framework is implemented on PyTorch (Paszke et al., 2019) with CUDA support and trained on one RTX 3090 GPU card. We use BPG (Bellard, 2014) as the

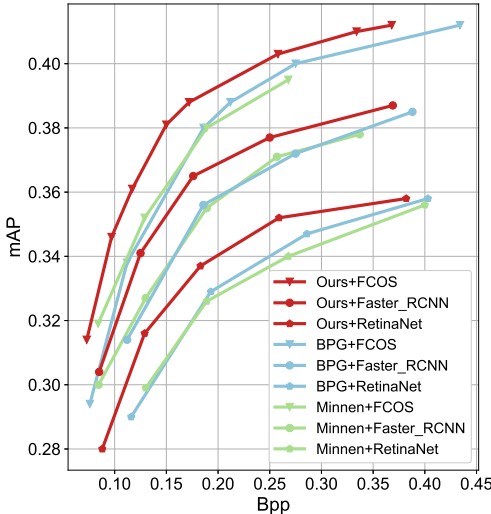 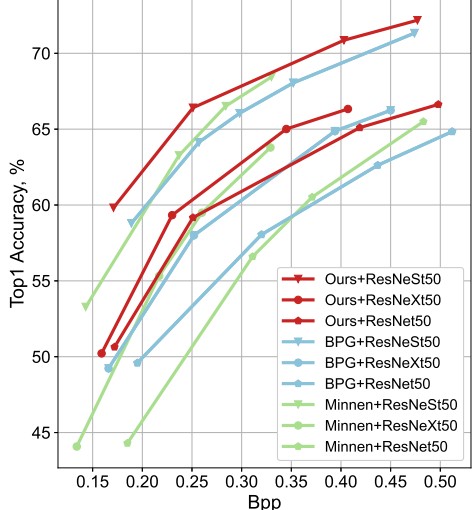

Figure 5: Rate-accuracy(mAP) curves from different compression methods for the object detection tasks on the COCO dataset.

Figure 6: Rate-accuracy(Top-1 accuracy) curves from different compression methods for the image classification task on the ImageNet dataset.

default traditional codec with different $QP$ values ($QP = 28, 31, 34, 37, 41$) and the corresponding $\lambda$s in Eq. 1 are set as $0.5, 1, 2, 4, 8$, empirically. The trade-off parameter $\beta$ is set as $0.5$. The weights of the downstream networks, like FCOS (Tian et al., 2019), remain fixed throughout the entire training process unless stated otherwise.

The whole training process has the following stages: First, based on the finetuning procedure in Section 3.3, we can obtain several proxy networks that mimic the BPG codec with different quantization parameters. Then we end-to-end optimize the neural preprocessing module without the quantization adaptive layers according to the loss function in Eq. 1 and set the $QP$ of the BPG codec to a fixed value, such as $QP = 34$. Finally, we add the quantization adaptive layers into the preprocessing module and further train the preprocessing module by randomly sampling $QP$ values. Specifically, we use the Adam optimizer (Kingma & Ba, 2014) and the initial learning rate is set as $1e-4$. The framework is optimized for 400k, 120k and 100k steps during the three training stages. And the learning rate is reduced to $1e-5$ after 320k, 80k and 60k steps when the loss becomes stable. The whole training process takes about five days.

## 4.2 MAIN RESULTS

We compare our preprocessing enhanced image compression method with existing traditional codec BPG (Bellard, 2014) and neural network based compression model (Minnen et al., 2018). BDBR (Bjontegaard, 2001) is used to measure compression performance in terms of the accuracy of the downstream tasks and the negative value represents the bitrate saving at the same accuracy. We use FCOS (Tian et al., 2019) and ResNet50 (He et al., 2016) as the default backbone networks for object detection and image classification and train the corresponding NPP modules, respectively.

**Object Detection** Fig. 5 shows the rate-accuracy curve from the different backbone networks and compression approaches on the COCO dataset. It is obvious that our preprocessing enhanced image compression method shows a much better rate-accuracy trade-off than the baseline approaches on the downstream object detection task. Specifically, compared with the existing BPG codec and learned compression model, our neural preprocessing enhanced codec saves 20.3% and 19.5% bitrate at the same mAP value when evaluated on FCOS, respectively. We further provide the visualization results in the Appendix.

**Image Classification** We also compare our method with the traditional and learning based codecs on the image classification task. Fig. 6 shows the rate-accuracy (Top-1) curves from different compression methods on the ImageNet dataset (Deng et al., 2009). It is noted that our approach still achieves better rate-accuracy performance and saves more than 22.0% bitrate when compared with traditional codec BPG (Bellard, 2014) by evaluating it on the ResNet50 (He et al., 2016) model.

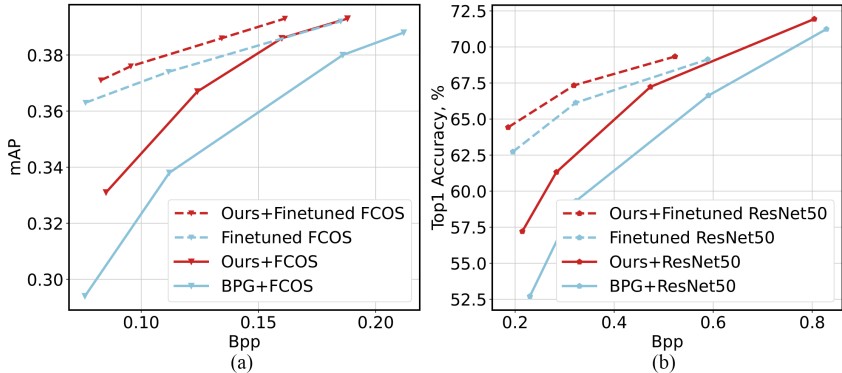

Figure 7: red(a) Rate-accuracy(mAP) curves of fine-tuned methods for the object detection tasks on COCO dataset. (b) Rate-accuracy curves of fine-tuned methods for the image classification task on ImageNet dataset.

Table 1: Inter-task Transfer BDBR(%) results of the Preprocessing Model

| Anchor Task(Backbone) | Transfer Task (Backbone) | | | |
|---|---|---|---|---|
| | Detection (FCOS) | Classification (ResNet50) | Segmentation (DeepLabV3) | Pose estimation (Deeppose) |
| Detection(FCOS) | — | -21.7 | -12.3 | -24.6 |
| Classification(ResNet50) | -13.9 | — | -12.6 | -14.4 |

## 4.3 GENERALIZATION ABILITY

To demonstrate the generalization ability of the optimized neural preprocessing module, we directly used the pretrained module for different downstream backbone networks, downstream tasks and traditional codecs.

**Different Backbones Networks.** For the object detection task, we perform new experiments by directly applying the NPP optimized for FCOS to other backbone networks like RetinaNet (Lin et al., 2017). Experiments in Fig. 5 show that our compression approach can still outperform the baseline methods and reduce 19.5% and 18.8% bitrate when compared with BPG on the downstream Faster-RCNN (Ren et al., 2015) and RetinaNet (Lin et al., 2017) models, respectively. Similar results are observed in Fig. 6 when we use the pre-trained NPP for ResNet50 to other backbones like ResNeSt and ResNeXt.

**Different Tasks.** Further, we transfer the pretrained NPP module optimized for one specific task to other three different vision tasks without fine-tuning. The results in Table. 1 show that our method is still useful and achieves significant bit saving. For example, we directly apply the NPP module optimized for detection to segmentation task and achieve 12.3% bitrate savings.

**Different Codecs.** Here, we also apply the NPP module optimized for the BPG codec to JPEG and VVC codec without fine-tuning. Experimental results show that our method can save about 10.0% bitrate for JPEG when evaluating on Faster-RCNN network in object detection, and can save about 15.6% bitrate for VVC when evaluating on the ResNet network in image classification, respectively. We present the figures of R-D curves in the Appendix.

## 4.4 ABLATION STUDY AND MODEL ANALYSIS

**Analysis of End-to-end Optimization.** In the proposed approach, we use BPG to generate reconstructed images in the forward propagation, while the gradients of the proxy network are used in the backward propagation to update the parameters of the preprocessing module. Here, we also provide the result of directly using the proxy network in both forward and backward propagation; however, we still use the BPG codec in the inference stage.

Experimental results show that this alternative solution (NPP+Minnen) can also effectively optimize the preprocessing module and improve the rate-accuracy performance. As shown in Fig. 8(a), com-

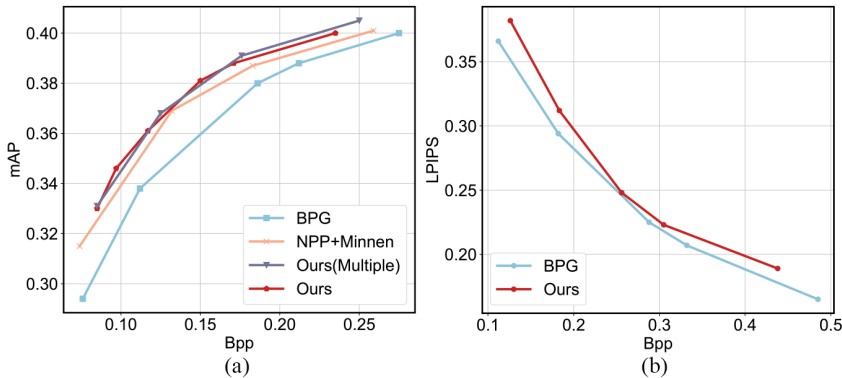

Figure 8: (a) Ablation study. (b) Rate-LPIPS curves of our approach and BPG on Kodak dataset.

pared with the original BPG codec, it saves about 14.6% bitrate at the same mAP value; however, our proposed training strategy is more effective and achieves 20.3% bitrate saving. The reason is that our approach uses BPG to generate the reconstructed images in the forward pass, which is consistent with the actual inference stage.

**Analysis of Quantization Adaptation Strategy.** Our proposed NPP module is quantization adaptive and can be used for BPG codec with different $QP$s. Here, we provide another alternative solution, *i.e.,* Ours(Multiple), where the quantization adaptive layers are removed and we train different NPP modules for different $QP$s in BPG. Experiments show that it has marginal improvements compared to our quantization adaptive implementation (See Fig. 8(a)). However, it needs to train and store multiple NPP models, which adds more storage burden to the encoder side.

**Experimental Results for Fine-tuned Downstream Networks** In some scenarios, the downstream task models can also be fine-tuned to further boost the performance for compressed images. To verify the effectiveness of our method in this case, we fine-tuned the target detection and classification network using BPG decoded images and trained our method for these new fine-tuned backbones. The experimental results in Fig. 7(a)(b) show our method still achieves 17.31% and 21.27% bitrate savings. Implementation details are provided in the Appendix.

**Compression Performance in terms of Human Visual System.** We also analyze the compression performance of our preprocessing enhanced image compression approach in terms of the human visual system.

Since our compression framework is optimized for machine vision tasks, the compression performance in terms of PSNR drops, which is no surprise. However, when we use more perceptual related metrics like LPIPS (Zhang et al., 2018), we found the gap is narrowing and our approach consumes an additional 8.5% bitrate when compared with the traditional baseline codec BPG in Fig. 8(c). More results are in the Appendix.

**Running Time and Complexity.** The number of parameters of our NPP module is 9.42M. For the input image with a size of $224 \times 224$, the inference time of our NPP module is only 4.23ms on a single RTX3090 GPU, suggesting that it brings little computational complexity to the existing pipeline.

## 5 CONCLUSION

In this work, based on traditional image compression algorithms, we propose a preprocessing enhanced image compression framework for downstream machine vision tasks. We introduce the neural preprocessing module to achieve a better trade-off between coding bitrate and the performance of machine vision tasks. Furthermore, we propose to use the proxy network to deal with the non-differentiable problem of the traditional codecs, which ensures that the gradients can be back-propagated to the neural preprocessing module and achieves the end-to-end optimization. Experiments show that our framework outperforms existing image codecs in several downstream tasks. More importantly, our approach shows strong generalization ability for different codecs, backbones, and tasks.

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

# A APPENDIX

## A.1 EFFECTIVENESS OF THE PROXY NETWORK.

To demonstrate the approximate ability of the proposed proxy network, we compared the rate-distortion performance between the BPG codec and the corresponding proxy network. The experimental results show that the proxy network has a similar performance to BPG (-1.4% BDBR gap), which indicates that it is feasible to replace the BPG codec in the training stage with the proxy network.

## A.2 VISUALIZATION OF DOWNSTREAM RESULTS

We provide the visualization results in Fig. 9 and it is evident that our neural preprocessing module is beneficial for the downstream tasks. For example, the reconstructed images produced by our method in the first and second row can be correctly classified while the corresponding result from BPG is wrong. At the same time, the proposed method also consumes less bitrate compared with BPG (0.42 vs. 0.49). We have a similar observation for the object detection task in the third and fourth row. The small objects can be recognized in our compressed results with less bitrate while they're missed in the results of BPG compressed images.

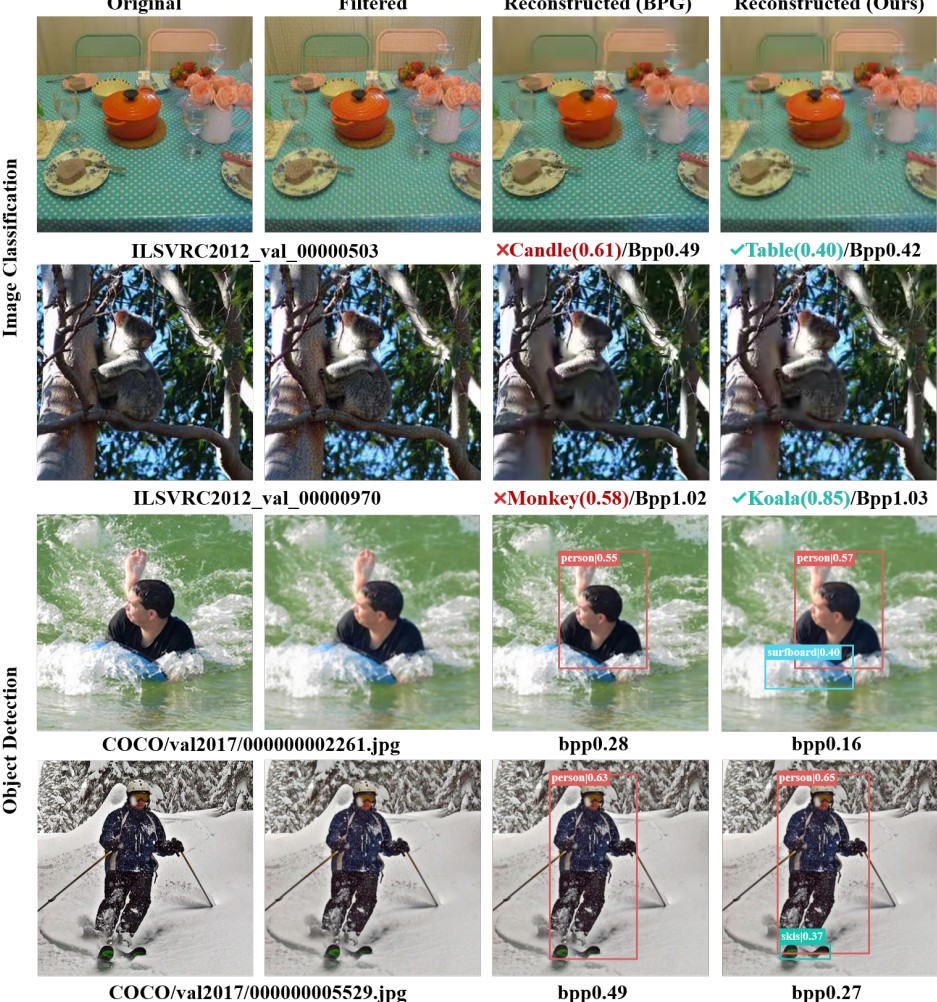

Figure 9: Visualization results of the downstream tasks.

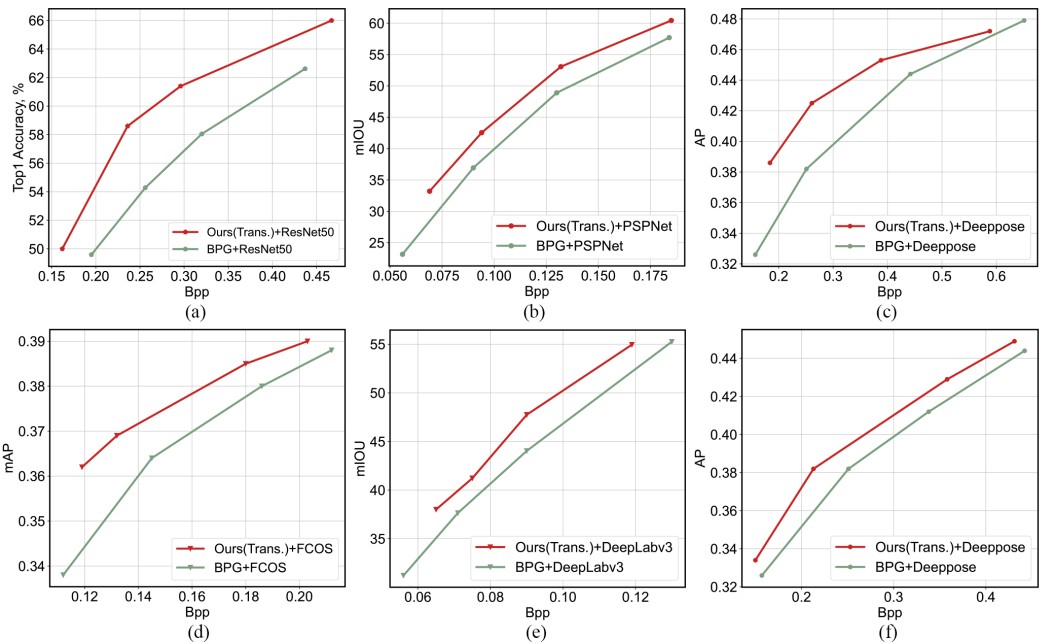

Figure 10: (a) Inter-task transfer of our NPP from object detection to image classification. (b) Inter-task transfer of our NPP from object detection to semantic segmentation. (c) Inter-task transfer of our NPP from object detection to pose estimation. (d) Inter-task transfer of our NPP from image classification to object detection. (e) Inter-task transfer of our NPP from image classification to semantic segmentation. (f) Inter-task transfer of our NPP from image classification to pose estimation.

### A.3 IMPLEMENTATION DETAILS OF FINE-TUNING DOWNSTREAM NETWORKS

In the scenarios outlined above, our focus is predominantly on the image compression component. We've operated under the assumption that the downstream visual task model remains static, without any fine-tuning or optimization for images that have undergone lossy compression. Nevertheless, there are instances where the downstream model can be fine-tuned to elevate the performance on compressed images. To demonstrate our method's efficacy under such circumstances, we fine-tuned target detection and image classification networks using BPG-decoded images, followed by training the neural preprocessing network for empirical validation. In particular, FCOS and ResNet were fine-tuned for 100,000 and 160,000 steps, respectively, across varying QPs, yielding notable improvements in bpp-accuracy performance. Subsequently, we integrated the NPP network, fine-tuning it further for 40,000 steps.

### A.4 MORE EXPERIMENTAL RESULTS FOR GENERALIZATION ABILITY

We provide the figures of R-D curves for applying NPP module optimized for BPG to JPEG and VVC codec, seperately in Fig. 11(a,b).

**Different Codecs.** We provide more experimental results for JPEG and VVC compression. Here, we apply the NPP module optimized for the BPG (Bellard, 2014) codec to the JPEG (Wallace, 1992) and VVC (Bross et al., 2021) without any fine-tuning. Experimental results in Fig. 11(a) show that our proposed preprocessing enhanced JPEG compression can save more than 8.5% and 10.0% bitrate compared with original JPEG codec when evaluating on the FCOS and Faster-RCNN backbone networks, respectively. Similar results are shown in Fig. 11(b) where our preprocessing enhanced VCC compression achieves more than 15.6% and 12.9% bitrate saving compared with the original VCC codec when evaluating on the ResNet and ResNeSt networks, respectively.

**Different Tasks.** To demonstrate the generalization ability of the preprocessing module in different tasks, we apply the preprocessing model optimized for the object detection task to the image

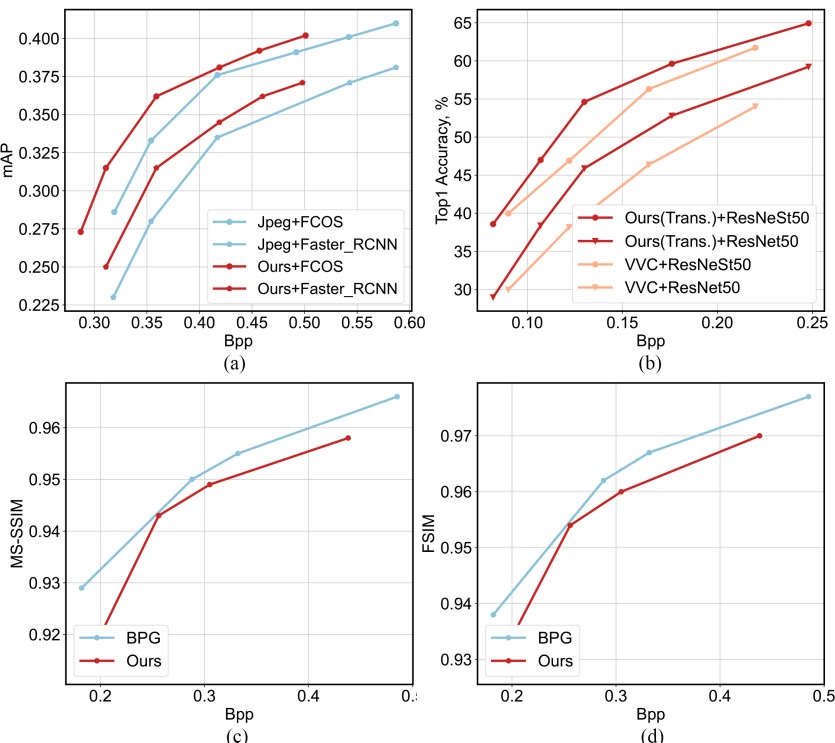

Figure 11: (a) Rate-mAP curves from our NPP enhanced JPEG codec and native JPEG codecs. (b) Rate-accuracy curves from our NPP enhanced VVC codec and native VVC codecs. (c) Rate-MSSSIM curves from ours approach and BPG codec on Kodak dataset. (d) Rate-FSIM curves from ours approach and BPG codec on Kodak dataset.

classification, segmentation and pose estimation tasks. The results in Fig. 10(a,b,c) show that our transfer method (Ours(Trans.)) is still useful and achieves more than 21.7%, 12.3% and 24.6% bitrate reduction, respectively. Additionally, We have the same observation in Fig. 10(d,e,f) when we apply the preprocessing model optimized for the image classification task to the object detection, segmentation and pose estimation tasks, which can save more than 13.9%, 12.6 % and 14.4% bitrate respectively. It is obvious that our model has a strong generalization ability between different machine vision tasks.

### A.5 MORE EXPERIMENTAL RESULTS FOR HUMAN VISUAL SYSTEM

We employed additional metrics, such as MS-SSIM (Wang et al., 2003) and FSIM (Zhang et al., 2011), to evaluate the performance of our preprocessing enhanced image compression approach in the context of the human visual system using the Kodak dataset. Relative to the conventional baseline codec BPG, our method incurs an 8.8% and 7.3% increase in bitrate, as illustrated in Fig. 11(c,d).

### A.6 ANALYZING THE EFFECTIVENESS OF NPP MODULE USING GRADCAM

We provide additional figures and employ GradCAM to evaluate the efficacy of the NPP module as shown in Fig. 12. In the first row, the classification model's attention is more expansive over the animal in the filtered image, evidenced by a pronounced red region in the heatmap, which may potentially benefit the model in making correct prediction. Consequently, after compression by BPG, the ResNet50 model accurately identifies the compressed filtered image as "Koala", whereas it misclassifies the compressed original image as "Monkey" at a comparable bpp of 1.03.

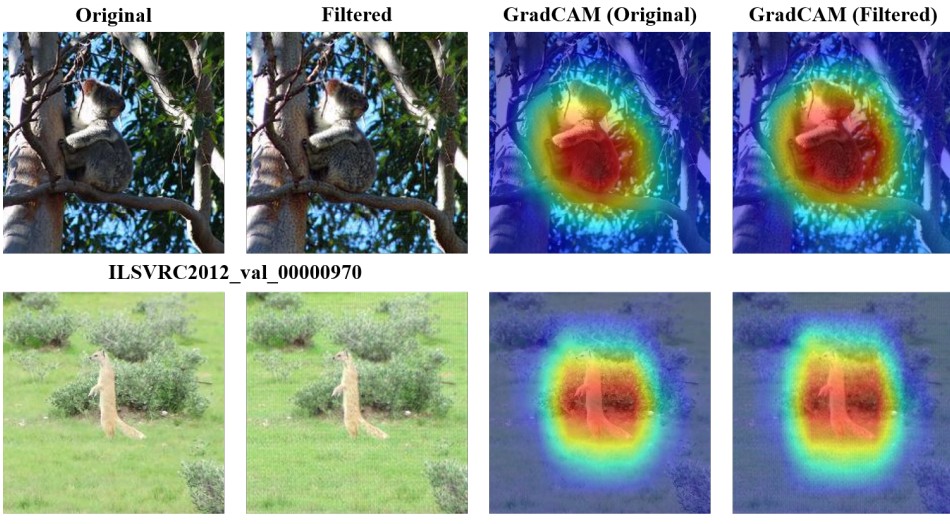

Figure 12: Visualization results of the downstream tasks.

A similar observation is noted in the second row. The ResNet50 classifier correctly identifies the compressed filtered image as "Mongoose" at bpp 0.30, but misclassifies the compressed original image as "Banded Gecko" at a slightly higher bpp of 0.37.

