# OpenReview forum: "Preprocessing Enhanced Image Compression for Machine Vision"
_ICLR.cc/2024/Conference — ICLR 2024 Conference Withdrawn Submission_

### Official Review · Reviewer_fHZ6 · 2023-10-25

**Soundness:** 2 fair
**Presentation:** 2 fair
**Contribution:** 1 poor
**Rating:** 3
**Confidence:** 5

**Summary:**

The paper proposes adapting an image on the encoder side such that when it is compressed with BPG, the result is good for machine perception tasks such as classification and detection. I.e., given an input x, the calculate x' = preprocess(x), then compress with BPG, get x'' = BPG(x'), and then do classification/detection on x''.

**Strengths:**

I like the method that approximates BPG ("Proxy Network"). The evaluation is rather broad (various methods and computer vision models).

**Weaknesses:**

I don't think the overall approach is well motivated, at least not as presented. There are two conflicting goals here: conserve image content after decompression, and be good in machine perception tasks. However, there is issues with both
A) The main text only reports LPIPS differences, and there are no PSNR numbers. From the visual examples we see that the method quite significantly alters the input image (eg Fig 4b).
B) From Fig 5 and 6 I'm seeing large drops in performance as we go down in bpp. It is unclear to me which tasks are fine with eg ~14% drops in accuracy (Fig 6, top red line).

What's concerning is that the preprocessing module is a deep neural network, that has to run on the sender. So, if machine perception is the goal, why not run the resnet on the sender, and encode the label? Eg classification with 1024 labels is only 10 bits!

While the proxy network is a cool way to approximate BPG, I just don't see the overall value proposition for an ICLR paper.

**Questions:**

What's the end goal here?

---

### Official Review · Reviewer_uQPP · 2023-10-27

**Soundness:** 2 fair
**Presentation:** 2 fair
**Contribution:** 2 fair
**Rating:** 3
**Confidence:** 5

**Summary:**

This work introduces a preprocessing enhanced image compression method designed to meet the growing demand of machine vision systems, improving the balance between image quality and compression efficiency for machine analysis tasks.

**Strengths:**

This paper sounds interesting, providing a solution to enhance the image quality for CV tasks. The paper is easy to read and understand.

**Weaknesses:**

However, the paper has many weaknesses, including low generation ability, practicality and insufficient experiments.

Generation ability:

1) Different codecs, tasks and backbones require training a new NPP module to adapt to them.

2) It's impossible to verify that this method is suitable for most kinds of images.

Practicality:

3) It can't be deployed to most devices to acquire the raw image data as shown in Figure 1 (b).

4) As described in the paper, NPP recovers images from low-quality compressed JPG images, which has no intrinsic difference from image enhancing.

5) As we know, image resolution has a great impact on CV tasks, such as classification or Detection, so whether the improvement of this method is better than post-processing enhancement (e.g.,  super-resolution).

Insufficient experiments:

6) The baselines or backbones used in CV tasks are not the latest, lacking comparison with methods after 2020, such as all kinds of Transformers.

7) It is vital to compare with the latest Learned Image Compression methods as mentioned the computational complexity is usually high, but I think NPP also has high computational complexity.

8) The proxy network may be a key contribution, but the paper has little context to reveal some insights about it.

**Questions:**

See in Weaknesses.

---

### Official Review · Reviewer_maca · 2023-10-31

**Soundness:** 3 good
**Presentation:** 3 good
**Contribution:** 3 good
**Rating:** 5
**Confidence:** 5

**Summary:**

The paper introduces a preprocessing-enhanced compression method that builds on traditional codecs, ensuring standard compatibility. It utilizes a neural preprocessing module to maintain semantic information for downstream tasks, while a proxy network is used to achieve back-propagation for traditional non-differential codecs.

**Strengths:**

1. The NPP module presented in the paper is plug-and-play, with a wide range of applicability.
2. The proxy network proposed in the paper effectively addresses the gradient backpropagation issues faced by traditional encoders.
3. The organization of the article is excellent, clear, and easy to understand.

**Weaknesses:**

1.	This method needs to be compared with some other relevant Image Coding for Machine methods, such as Omni-ICM [1] and TransTIC [2], to demonstrate its effectiveness.
[1] Feng R, Jin X, Guo Z, et al. Image coding for machines with omnipotent feature learning[C]//European Conference on Computer Vision. Cham: Springer Nature Switzerland, 2022: 510-528.
[2] Chen Y H, Weng Y C, Kao C H, et al. TransTIC: Transferring Transformer-based Image Compression from Human Perception to Machine Perception[C]//Proceedings of the IEEE/CVF International Conference on Computer Vision. 2023: 23297-23307.
2.	Data with an upper bound needs to be added in Figure5,6 (i.e., results from inputting uncompressed images to the task network) to observe the method's maximum potential.
3.	Using a proxy network to mimic the BPG codec, different lambdas are required to train for different QPs. The paper only mentioned choosing lambda mainly based on experience. It needs a more detailed explanation of how to quickly select an appropriate lambda, as well as the impact of different lambda values on the results of the same QP.

**Questions:**

Please refer weakness part